# Genomic Epidemiology Identifies Azole Resistance Due to TR_34_/L98H in European *Aspergillus fumigatus* Causing COVID-19-Associated Pulmonary Aspergillosis

**DOI:** 10.3390/jof9111104

**Published:** 2023-11-13

**Authors:** Benjamin C. Simmons, Johanna Rhodes, Thomas R. Rogers, Paul E. Verweij, Alireza Abdolrasouli, Silke Schelenz, Samuel J. Hemmings, Alida Fe Talento, Auveen Griffin, Mary Mansfield, David Sheehan, Thijs Bosch, Matthew C. Fisher

**Affiliations:** 1Medical Research Council Centre for Global Infectious Disease Analysis, Imperial College London, London W2 1PG, UK; johanna.rhodes@imperial.ac.uk (J.R.); s.hemmings@imperial.ac.uk (S.J.H.); matthew.fisher@imperial.ac.uk (M.C.F.); 2UK Health Security Agency, London EP14 4PU, UK; 3Department of Medical Microbiology, Radboudumc Center for Infectious Diseases (RCI), Radboud University Medical Center, P.O. Box 9101, 6500 HB Nijmegen, The Netherlands; paul.verweij@radboudumc.nl; 4Department of Clinical Microbiology, St. James’ Hospital Campus, Trinity College Dublin, D08 NHY1 Dublin, Ireland; rogerstr@tcd.ie (T.R.R.); talenta@tcd.ie (A.F.T.); mansfime@tcd.ie (M.M.); sheehad3@tcd.ie (D.S.); 5Radboudumc-CWZ Center of Expertise for Mycology, Radboudumc Center for Infectious Diseases (RCI), Radboud University Medical Center, P.O. Box 9101, 6500 HB Nijmegen, The Netherlands; 6Center for Infectious Disease Research, Diagnostics and Laboratory Surveillance, National for Public Health and the Environment (RIVM), P.O. Box 1, 3720 BA Bilthoven, The Netherlands; thijs.bosch@rivm.nl; 7Department of Infectious Diseases, Imperial College London, London W2 1NY, UK; alireza.abdolrasouli@nhs.net; 8Department of Infectious Diseases, King’s College Hospital, London SE5 9RS, UK; 9Infection Sciences, King’s College Hospital, London SE5 9RS, UK; silke.schelenz@nhs.net; 10School of Immunology & Microbial Sciences, King’s College London, London WC2R 2LS, UK; 11Department of Microbiology, Our Lady of Lourdes Hospital, A92 VW28 Drogheda, Ireland; 12Department of Microbiology, Royal College of Surgeons, D02 YN77 Dublin, Ireland; 13Department of Microbiology, St. James’ Hospital, D08 NHY1 Dublin, Ireland; augriffin@stjames.ie

**Keywords:** *Aspergillus fumigatus*, azole-resistant *Aspergillus fumigatus*, COVID-19-associated pulmonary aspergillosis, CAPA, coinfection, genetic epidemiology, genomic analysis

## Abstract

*Aspergillus fumigatus* has been found to coinfect patients with severe SARS-CoV-2 virus infection, leading to COVID-19-associated pulmonary aspergillosis (CAPA). The CAPA all-cause mortality rate is approximately 50% and may be complicated by azole resistance. Genomic epidemiology can help shed light on the genetics of *A. fumigatus* causing CAPA, including the prevalence of resistance-associated alleles. We present a population genomic analysis of 21 CAPA isolates from four European countries with these isolates compared against 240 non-CAPA *A. fumigatus* isolates from a wider population. Bioinformatic analysis and antifungal susceptibility testing were performed to quantify resistance and identify possible genetically encoded azole-resistant mechanisms. The phylogenetic analysis of the 21 CAPA isolates showed that they were representative of the wider *A. fumigatus* population with no obvious clustering. The prevalence of phenotypic azole resistance in CAPA was 14.3% (*n* = 3/21); all three CAPA isolates contained a known resistance-associated *cyp51A* polymorphism. The relatively high prevalence of azole resistance alleles that we document poses a probable threat to treatment success rates, warranting the enhanced surveillance of *A. fumigatus* genotypes in these patients. Furthermore, potential changes to antifungal first-line treatment guidelines may be needed to improve patient outcomes when CAPA is suspected.

## 1. Introduction

*Aspergillus fumigatus* is the predominant causative agent of invasive aspergillosis (IA) [1,2], with an estimated annual prevalence of > 250,000 cases globally and a mortality rate of 30 to 95% [3,4]. Normally, only those who are immunocompromised or immunosuppressed are highly susceptible to developing IA [2]; however, there is a widening group of patients who are at risk of IA, including those with severe influenza or COVID-19.

In 2019–2020, the novel virus severe acute respiratory syndrome coronavirus 2 (SARS-CoV-2), causing COVID-19 disease, spread across the globe. Opportunistic pathogens have been widely reported as causing secondary infections in COVID-19 patients with lung damage and a notable proportion of these are fungal coinfections (12.6%) [5]. A large proportion of fungal coinfections in COVID-19 cases are caused by *Aspergillus* species, including *A. fumigatus*, giving rise to COVID-19-associated pulmonary aspergillosis (CAPA) [6,7]. The European Confederation for Medical Mycology (ECMM) and the International Society for Human and Animal Mycology (ISHAM) developed a consensus definition for CAPA at the end of 2020, stating that CAPA is a form of IA that is “in temporal proximity to a preceding SARS-CoV-2 infection” [7]. The global prevalence of CAPA is reported to be between 3.8 to 40%, with 15.1% of ICU-admitted COVID-19 patients fulfilling the ECMM definition of CAPA [6,7,8,9,10]. Furthermore, CAPA is characterised by low survival, with mortality ranging from 44 to 75% [10,11,12]. The MYCOVID cohort study showed that COVID-19 patients who received intensive care treatment and positive sputum cultures for *A. fumigatus* but could not be classified as CAPA had higher mortality (45.8%) than those who had negative *A. fumigatus* cultures (32.1%) [10]. However, this was lower than the group who had CAPA (61.8%) [10]. Therefore, so-called colonisation may not be as harmless as previously thought.

In recent years, *A. fumigatus* resistance to the azole antifungal drug class has emerged, with its prevalence rapidly increasing, and recently has been declared a public health issue [13,14]. In the environment, population genomic studies have determined that depending on the country, between 2.2 to 20% of isolates are azole-resistant, and this proportion is as high as 95.2% in Vietnam [15,16,17,18,19,20]. A cause for concern is the increase in the proportion of azole-resistant *A. fumigatus* (AR*Af*) identified in patients, as this is associated with treatment failure, increased mortality rates, and a doubling of health care costs [13,14]. In CAPA, the prevalence of azole resistance has not yet been established, as the sample size of the studies has been too small. Two CAPA genomic epidemiology studies have reported no resistant isolates [21,22] while one German study reported 22.2% (*n* = 6/27) [23] as resistant. Using the Clinical and Laboratory Standards Institute (CLSI) method, a second German study (*n* = 4) did not find any resistant isolates [24]. Finally, a Portuguese study reported a high prevalence of AR*Af* of 45.5% (*n* = 5/11 [25].

Genomic epidemiological methods have played a key role in delineating the genetic basis of AR*Af* [13,26,27]. The primary locus, with a high-frequency of non-synonymous SNPs (nsSNPs), known to be involved in azole resistance is the *cyp51A* gene encoding the 14α-sterol demethylase enzyme [13,26]. Common amino acid substitution hotspots are G54, L98H, G138, M220, and G448 (Table A1). Point mutations occur in isolates cultured from patients who have had long-term exposure to azole therapy [26]. Additionally, several tandem repeats (TRs) in the gene promoter region of *cyp51A* lead to the overexpression of the gene and are commonly associated with point mutations in the *cyp51A* gene (Table A1). The genotypes of *A. fumigatus* that contain TR-mediated resistance are more common in environmental isolates and isolates obtained from azole-naïve patients. However, there is increasing awareness that resistance mechanisms in AR*Af* are complex and involve multiple genes (Table A1) [16,26,27,28,29,30,31,32]. In studies that identified AR*Af* in CAPA isolates, Kirchoff et al. [23] discovered only one CAPA isolate that contained TR_34_/L98H and five polymorphisms of AR*Af* via a non-*cyp51A* mechanism.

Whilst the development of a clinical definition for CAPA has aided clinicians in its diagnosis and ensuring treatment is commenced in a timely manner to improve patient outcomes, important questions remain as to the genetic characteristics and identity of *A. fumigatus* causing CAPA and the prevalence of antifungal resistance [13,21,22,23,24,33]. In *A. fumigatus*, genomic epidemiological methods have begun to unravel the genetics of environmental and clinical antifungal resistance and the potential mechanisms of dispersion [1,13,16,32]. To date, there are only five genomic epidemiologic studies of CAPA using multiple methods [21,22,23,24,25]. The results of these analyses were inconclusive on the genetic and epidemiological relatedness of *A. fumigatus* causing CAPA. The current study is the largest transnational genomic epidemiological investigation to date of CAPA isolates to determine where the genotypes of CAPA isolates group in the wider *A. fumigatus* population. Secondly, this study aimed to identify the mechanisms and determine the frequency of azole-resistant polymorphisms in CAPA isolates.

## 2. Materials and Methods

### 2.1. CAPA Definition

The definition of CAPA was based on the 2020 ECMM/ISHAM consensus criteria [7]. In summary, patients must have a positive SARS-CoV-2 polymerase chain reaction (PCR), require intensive care for COVID-19 and have signs of invasive pulmonary aspergillosis (IPA) infection. Ideally, IPA is confirmed through the histopathological or direct microscopic detection of fungal hyphae obtained by lung biopsy, therefore, showing signs of tissue invasion and/or damage [7,34]. However, samples from bronchoalveolar lavage, bronchial aspirate, and tracheal aspirate were used as alternatives, as lung biopsies could rarely be performed in CAPA patients [7]. A positive SARS-CoV-2 PCR test must occur between two weeks prior to hospital admission and up to 96 h after ICU admission. CAPA may be further categorised depending on the sensitivity of the diagnostic method used. That is, CAPA can be either proven, probable, or possible [7,11]. A fourth category was added in cases where *A. fumigatus* was isolated from the sputum of a patient with COVID-19 receiving intensive care, but there were no clear signs of invasive disease [10]. Furthermore, there was no follow-up bronchoscopy to determine if there was an invasive disease. Therefore, for the first analysis, the patients were categorised as non-CAPA and only colonised with *A. fumigatus* (‘colonising’). In the second analysis, the colonising isolates were categorised as clinical non-CAPA isolates.

### 2.2. Fungal Isolates

Twenty-one CAPA isolates from four European countries between 2020–2021 were included (Table 1). In this study, all CAPA isolates were recovered as per the ‘standard of care’ based on the ECMM criteria [7]. Four CAPA isolates (CAPA-A–D) were from four separate cases that originated from two Cologne hospitals in Germany [24,35]. Six CAPA isolates were from London, UK (C422–C425, C611, C612). A further two isolates originated from two hospitals in the Netherlands (C403 and C408). A further nine isolates were from two hospitals in Dublin, Ireland (C434–C441, C444). Isolate C444 has previously been described in a case report [36].

In this study, the genetic relatedness of CAPA isolates was compared to (1) *A. fumigatus* isolates from non-CAPA IA cases, and isolates colonising non-CAPA patients; and (2) non-CAPA clinical and environmental isolates from the wider *A. fumigatus* population. The first analysis involved comparing CAPA isolates to two other groups. The first group comprised twelve *A. fumigatus* isolates from eleven patients with different types of IA (C120, C137–C140, C143, C307, C323, C360, C372, C376, C442). This included four isolates from a patient with necrotising aspergillosis, one trauma patient, two allergic bronchopulmonary aspergillosis (ABPA) patients, two with ABPA and asthma, and one with chronic pulmonary aspergillosis (CPA), and two with IPA (Table A2). The selection criteria for these IA isolates were based upon the European Organization for Research and Treatment of Cancer and the Mycoses Study Group definition for invasive disease [34], and were from the UK and Ireland. The second group comprised eight ‘colonising’ isolates: seven from three Dutch Hospitals (C402, C404–C407, C409, C410) and one from a patient in Ireland (C443) [32,37,38]. The second analysis involved comparing 21 CAPA, 167 non-CAPA clinical, and 73 environmental *A. fumigatus* isolates. In summary, 218 UK and Irish *A. fumigatus* isolates that had been analysed by Rhodes et al. [32] and 23 additional *A. fumigatus* isolates (C307, C323, C360, C372, C376, C402, C404–C407, C409, C410, C426–C433, C442, C443) were used (Table A2). All 261 isolates can be found in the Microreact project [37] at https://microreact.org/project/mPbPTWS2jTvvdNGmuDVyia-capa (accessed on 4 November 2023).

#### Clinical Characterisation of the CAPA Isolates

Applying the ECMM/ISHAM criteria, sixteen isolates were from probable CAPA cases (C422–C425, C434–C441, C444, C611, CAPA-C, and CAPA-D) and five were from possible CAPA cases (C403, C408, C612, CAPA-A, and CAPA-B).

### 2.3. Antifungal Susceptibility Testing

Susceptibility to azole antifungal agents was carried out on all 21 of the CAPA isolates within this study. Isolates were first screened using the low-cost tebuconazole screening test (Tebucheck) to determine which isolates were resistant and then VIPcheck^TM^ was used to give more information on which medical azoles the isolates were resistant to [39]. Fifteen of the CAPA isolates were prepared as per the protocol set out in Brackin et al. [39]. Briefly, isolates were inoculated in 25 cm^3^ tissue culture flasks containing Sabouraud dextrose agar and culture for a minimum of 2 days at 45 °C. Conidia were collected in sterilised 0.05% (*v*/*v*) Tween-80 (Calbiochem^®^, Sigma-Aldrich, Gillingham, UK using filtration through sterilised glass-wool. A final standardised spore suspension was achieved with a cell density reading at 600 nm between 0.09 and 0.12 using sterilised 0.05% (*v*/*v*) Tween-80 with a standard spectrophotometer [39]. Twenty microlitres of final standardised spore suspensions were inoculated into Tebucheck multi-well and VIPcheck^TM^ (Mediaproducts BV, Groningen, NL) plates. The Tebucheck multi-well plate consists of four wells comprising of drug-free (growth control well), 6 mg/L, 8 mg/L, and 16 mg/L of tebuconazole [39]. All Tebucheck multi-well plates were made in-house as per the protocol set out in Brackin et al. [39]. The VIPcheck^TM^ multi-well plate consists of wells with 4 mg/L itraconazole, 2 mg/L voriconazole, 0.5 mg/L posaconazole, and a drug-free growth control well [40]. Due to a lower concentration of spore suspension used, cultured plates were left at 45 °C to ensure there was complete growth in the control wells (48 h). Each strain was compared to a non-CAPA clinical isolate C154, which was EUCAST susceptible and verified by both Tebucheck and VIPcheck^TM^.

The scoring of Tebucheck plates is as follows: a well that had no growth (0 to <10% of the well) was scored 0; if there was partial growth (10 to 50% growth) the well was scored 0.5; and if there was full growth (50 to 100%) then the well was scored 1 [39]. Any isolates with a score greater than 1 were considered resistant to tebuconazole [39].

The scoring of VIPcheck^TM^ can be found in Buil et al. [40]. Briefly, after 24 h, if there was uninhibited growth in any well which contained an azole, then the well would be scored 2 and the isolate would be resistant to that azole. If there was minimal growth, then this would be scored 1 and the isolate would therefore be partly resistant. If there was no growth in the azole-containing wells, then the well would be scored a 0 and the isolate would be considered susceptible to the azole.

Antifungal susceptibility data from previously published research was included in this study. The isolates C1–C200, E9–E206, and U1-3 [32]; C307, C323, C360, C372, and C376 [38]; and C444 [36] susceptibility testing was performed as per the protocol set out in Rhodes et al. [32], based upon the recent EUCAST methodology [41]. The Dutch isolates (C403–C410) had susceptibility testing performed using EUCAST methodology [41]. Finally, azole-resistant minimum inhibitory concentrations (MICs) were measured on two Irish isolates (C438 and C441) and the previously published CAPA-A to D [24], using CLSI broth dilution technique M38-A2 by the Mycology Reference Laboratory Bristol UK [42]. A caveat of the CLSI method is that it has different breakpoint values compared to the EUCAST method [41,42].

### 2.4. Genomic DNA Preparation and Whole Genome Sequencing

Whole-genome libraries of the *A. fumigatus* isolates were prepared and sequenced by different research teams and at different times; thus, different platforms were used. Extraction of genomic DNA from the Irish and UK CAPA isolates (*n* = 15) and some additional Irish non-CAPA isolates (clinical, *n* = 2, and environmental, *n* = 8) was carried out at Imperial College London. Briefly, gDNA was extracted using the MasterPure^TM^ Complete DNA and RNA Purification Kit (Lucigen, Middleton, USA), including an additional bead-beating step using a FastPrep-24^TM^. Extracted gDNA was purified using a DNeasy Blood and Tissue Kit (Qiagen, Germany), with the concentration measured using a Qubit fluorometer and dsDNA Broad Range Assay kit (Invitrogen, ThermoFisher Scientific, Karnataka, India). DNA purity was assessed using NanoDrop^TM^ spectrophotometry. Purified gDNA was stored at −20 °C prior to the construction of gDNA libraries, normalisation, and indexing (Earlham Institute, Norwich, UK). Libraries were run on a NovaSeq 6000 SP v1.5 flow cell (Illumina, Cambridge, UK) to generate 150 bp paired-end reads.

CAPA (C403, C408) and colonising (clinical non-CAPA) (C402, C404–C407, C409, C410) isolates from the Netherlands (*n* = 9) were sequenced using NextSeq 550 sequencer (Illumina), and 218 UK and Irish isolates and the five IA isolates (C307, C323, C360, C372, C376) were all sequenced using a HiSeq2500 (Illumina) sequencer [32]. All these isolates generated 50-150bp paired-end reads. The four CAPA isolates obtained from Germany were sequenced from single-stranded circular DNA (ssCir DNA) using the MGISEQ2000 [24,43].

### 2.5. Bioinformatics Analysis

Raw Illumina whole-genome sequence (WGS) reads were quality checked using FastQC v0.11.9 (Brabham institute) and subsequently aligned to the Af293 reference genome using the Burrow-Wheeler Aligner alignment tool with maximal exact methods algorithm [44,45]. The quality of the alignment was improved and converted to sorted BAM format using sequence alignment/map (SAM) tools v1.15. To minimise inaccurate identification of single-nucleotide polymorphisms (SNPs) due to PCR duplication error, duplicated reads were marked using Picard v2.18.7. SNPs were identified using ‘Haplotype Caller’ from the Genome Analysis Toolkit v4.0. To ensure high confidence calls, SNPs had to achieve at least 1 parameter, QD < 2.0, fisher strand > 60.0, mapping quality < 40.0, mapping quality rank sum test < −12.5, read positive rank sum test < −8.0, and SOR > 4.0. The above expressions have been rigorously tested and benchmarked [32]. SNPs were mapped to genes using vcf-annotator v0.5 (Broad Institute).

### 2.6. A. fumigatus and MAT Identification

Previously, 223 UK and Irish *A. fumigatus* isolates have been confirmed to be members of the *A. fumigatus* species complex using molecular methods [32]. In this study, the CAPA isolates were confirmed as *A. fumigatus* for congruency with Af293 using the nucleotide sequence of the calmodulin (*CaM*) gene from Af293, utilising the basic local alignment searching tool (BLAST, v2.13.0) from the National Centre of Biotechnology Information (NCBI) [46,47]. A CAPA isolate was identified as *A. fumigatus* if the top BLAST hit had a percentage identification of >99% [48].

The mating type of all 261 *A. fumigatus* isolates was identified using BLAST v2.13.0 (NCBI), using sequences from Pyrzak et al. [49].

### 2.7. Phylogenetic and Spatial Analyses

Whole-genome SNP data were converted to the presence/absence of an SNP with respect to the reference. SNPs identified as low confidence during variant filtration were assigned as missing. In the first analysis, as there were fewer than 50 taxa, maximum likelihood (ML) phylogenies were constructed using the gamma model (GTRGAMMA) of rate heterogeneity and rapid bootstraps over 1000 replicates in RAxML v8.2.9 (Stematakis, 2006 RAxML-VI-HPC) [50]. In the second analysis, with more than 50 taxa, maximum likelihood (ML) phylogenies were constructed using the CAT approximation (GTRCAT) of rate heterogeneity and rapid bootstraps over 1000 replicates in RAxML v8.2.9 (Stematakis, 2006 RAxML-VI-HPC) [50]. Phylogenies were annotated and visualised in ggtree and ggtreeExtra in R (v4.2.2).

Genetic similarity and population allocation were investigated via principal component analysis (PCA) and discriminant analysis of principal components (DAPC) based on whole genome SNP data in R (v4.2.2). DAPC was performed using adegenet [51]. To determine the number of principal components (PCs) to retain, the *a*-score method in the adegenet package was used. As the loadings of the PCs themselves are generally uninformative, to gain insight into how each isolate contributes to the cluster’s composition, compoplots (adegenet) were generated.

### 2.8. Statistical Analyses

Descriptive analysis of data was conducted in R v4.2.2 and Microsoft Excel. Chi (χ^2^)-test was performed to address the association between CAPA and azole-drug resistance. Significant value was considered for *p*-values < 0.05. Yates’s continuity correction was used to calculate χ^2^ in cases when values for a variable were less than 20.

## 3. Results

### 3.1. WGS of 21 CAPA Isolates

The reference-guided methods were used to analyse the genetic diversity of twenty-one CAPA isolates from four European countries (Table 1). All the isolates were confirmed to be *A. fumigatus*, sharing 100% similarity with the calmodulin (*CaM*) gene of the Af293 genome. All the sequenced genomes mapped > 96.2% (mean 97.8%) to the reference genome Af293 (Table 1), with a mean coverage of 151× (128–167×) for the German and Dutch CAPA isolates, and 42.1× (27.6–56.2×) for the UK and Irish isolates, reflecting the different sequencing methods and platforms used. The normalised whole-genome depth of coverage confirmed an absence of aneuploidy events in the CAPA isolates. However, deletions and duplications were observed; all the isolates observed had an approximately 260 kilo-base pairs (Kbp) deletion in Chromosome IV (Figure A1). This phenomenon has been observed in previous WGS studies, and the region of the rRNA repeat cluster may be included in this deletion [44,52]. Deletions were also found in Chromosome I (region of > 200 Kb), VII (region of > 300 kb), and VIII (approx. 60 Kb region), in 7 (24.1%), 14 (48.3%), and 8 (27.5%) isolates, respectively (Figure A1). These regions have not been identified as having any known importance in the context of azole resistance. Multiple peaks were observed in all the chromosomes from each isolate, which relate to the presence of copy number variations (CNVs).

An even proportion of mating idiomorphs in the CAPA isolates was observed, with 52% (*n* = 11/21) containing the *MAT1-1* gene and 48% (*n* = 10/21) containing the *MAT1-2* gene. This is consistent with what is expected in the wild, and is suggestive of a sexually recombining population [49,53]. Combining the CAPA isolates with the UK and Ireland *A. fumigatus* isolates, there was a bias towards the *MAT1-2* idiomorph (60.9%), which was statistically significant (χ^2^-test, *p*-value = 0.014, degrees of freedom, d.f. = 2). This means that the population may be more likely to reproduce asexually [32,49,53]. Although, there was no significant difference between source types (Table A3).

On average, the CAPA isolates differed from each other by 26,706 SNPs. There was only one CAPA isolate pair (C439–C440) that was highly similar (< 2671 SNPs); however, these were obtained from different patients. When the CAPA isolates were compared to the IA and colonised isolates, they differed by 30,986 SNPs.

### 3.2. Azole Resistance within CAPA Isolates Primarily Centred on Known Polymorphisms within cyp51A

Three CAPA (C438, C441, C444) isolates obtained from different locations in Ireland contained azole-resistant polymorphisms (TR_34_/L98H) (Table 2, Table A4, and Table A5). No known drug-resistant polymorphisms were identified in the Dutch, UK, or German CAPA isolates.

Susceptibility testing was performed to confirm the isolates containing azole-resistant polymorphisms have raised MICs to azole drugs. Of the CAPA isolates, three had scores of greater than 1 in VIPcheck^TM^ and/or Tebucheck (C438, C441 and C444) (Table 2). Thus, three of the 21 CAPA isolates were AR*Af*. The remaining CAPA isolates scored 1 in VIPcheck^TM^ and/or Tebucheck, and, therefore, are susceptible phenotypically. These results were verified independently using EUCAST and the CLSI broth microdilution methods (Table A4 and Table A5) [24,32,36].

### 3.3. Phylogenetic and Spatial Analysis Shows CAPA Isolates Are Highly Related to Non-CAPA Clinical and Environmental A. fumigatus

To test the phylogenetic relatedness of *A. fumigatus* cultured from CAPA patients with the existing collection of *A. fumigatus* from IA and colonised patients, the phylogenetic analysis of twenty-one CAPA, twelve non-CAPA IA isolates, and eight non-CAPA colonising isolates from four European countries was conducted (Table A2). This analysis revealed two broadly divergent clades, which are referred to as clade A and clade B, as previously reported (Figure 1 and Figure A2) [32,54]. Of the CAPA isolates, the majority (*n* = 15, 71.4%) were in clade B, and the remainder were in clade A (*n* = 6, 28.6%) (Figure 1 and Table A3). All CAPA AR*Af* isolates (C438, C441, C444) were situated in clade A. All the Dutch CAPA isolates were from clade B, whereas only 55.6% of the Irish CAPA isolates were from clade B. The UK and German CAPA isolates had 84.3% and 75% of isolates in clade B, respectively. Of the IA isolates (C120, C137–C140, C143, C307, C323, C360, C372, C376, C442), eight (66.6%) were in clade A and four (33.3%) were in clade B. The IA cohort contained ten AR*Af* isolates, which all contained *cyp51A* polymorphisms. The eight IA isolates in clade A were AR*Af* and two (C120 and C372) of the four IA isolates in clade B were AR*Af* isolates (Figure 1 and Appendix Table A4 and Table A5). Finally, seven of the eight colonising isolates comprised clade B and all the isolates where not AR*Af*.

Multivariate methods were used to identify and describe the genetically related clusters. The principal component analysis (four principal components, PCs) identified that the CAPA isolates come from the same population as the IA isolates (Figure 2A). Interestingly, the colonising isolates formed a subsection of the CAPA isolates (Figure 2A). The discriminant analysis of principal components (DAPC) and composition plot (five PCs) confirmed this observation (Figure 2B,C). All the isolate genotypes contained a mixture of CAPA, IA, and coloniser, with twenty-one of the isolates containing >20% CAPA genome (Figure 2B,C). Despite this, the isolates labelled as IA or coloniser are not genetically distinct from the CAPA isolates.

To gain a more in-depth understanding of the genetic relatedness of the CAPA isolates and whether they represent a distinct genetic identity or are drawn from the wider population of *A. fumigatus*, the phylogenetic relationship of the CAPA isolates with the wider *A. fumigatus* population was investigated. A phylogenetic analysis was conducted on 240 clinical and environmental isolates and twenty-one CAPA isolates (Figure A4). Again, two broadly divergent clades were observed: clades A and B (Figure 3, Figure A3 and Figure A4).

In total, 137 isolates (52.5%) lay within clade A and 124 isolates (47.5%) were from clade B. As seen in the first analysis, the CAPA isolates were largely associated with clade B (*n* = 15, 71.4%) (Figure 3). The proportions of each isolate (CAPA, non-CAPA clinical, and environmental) source type were associated with different clades, which were significantly different (χ^2^-test, *p* < 0.001; d.f. = 2) (Table A3). Additionally, the isolates containing azole-resistant polymorphisms were significantly associated with clade A (χ^2^-test, *p* < 0.001; d.f. = 1).

A multivariate analysis of the underlying genomic structure of the isolates confirmed that the genomes of the CAPA isolates could not be distinguished from the non-CAPA clinical and the environmental *A. fumigatus* isolates originating from the UK and Ireland (Figure 4).

## 4. Discussion

The COVID-19 pandemic has created a large and growing global cohort of patients at risk of developing *A. fumigatus* coinfections. Additionally, CAPA has higher mortality than COVID-19 itself (16 to 25% excess mortality rate) [7]. This, in part, has been due to the challenges of determining whether COVID-19 patients have IA [7,10]. Therefore, delays in commencing prompt antifungal treatment may have led to higher mortality rates [55]. Furthermore, WGS could address these issues by providing information on potential common sources, the genetic relatedness of the CAPA isolates with the wider *A. fumigatus* population, and the presence of AR*Af*, informing clinicians about the most effective way to treat CAPA patients [1,13]. In this study, WGS was carried out on twenty-one CAPA isolates from four European countries to explore the genomic epidemiology of *A. fumigatus* causing CAPA.

The present genomic analysis of CAPA isolates yielded three major findings. First, the CAPA isolates comprised a diverse range of *A. fumigatus* genotypes. Secondly, the CAPA genomes are an even mix of *A. fumigatus* genotypes found in previous studies in either the clinical environment or the environment. Finally, our study provides evidence of AR*Af* in at-risk patient cohorts. However, an important caveat is that the only AR*Af* found were from one country, likely due to the overall small sample size of *A. fumigatus* studied.

To gain insight into CAPA’s genetic and epidemiological relatedness, twenty-one CAPA isolates were compared to the different types of *A. fumigatus* samples including clinical non-CAPA and environmental isolates. This analysis provided evidence that the genotypes of the *A. fumigatus* isolates from the CAPA patients are genetically diverse, with two broadly divergent clades: A and B. A proportion of the CAPA isolates clustered in clade B (*n* = 15, 71.4%). In the sample of the IA isolates, 64% were in clade A. The phenomenon of the two-clade structure of *A. fumigatus* species was first identified in a large global genetic epidemiology study of over 4000 *A. fumigatus* isolates and has since been replicated in different *A. fumigatus* populations [32,54,56].

Even though the genotypes of the CAPA isolates in this study were biased towards clade B, the isolates were genetically diverse. The genetic relatedness between the CAPA isolates is on average 26,706 SNPs. The genetic relatedness between the isolates was higher when compared across the twelve IA and eight colonising isolates (average 30,986 SNPs) and the wider *A. fumigatus* population (25,448 SNPs), comparable to previous studies [32,56]. Only one pair of isolates was highly related (<2671 SNPs; C439–C440) and originated from the same country, but were from different patients, suggesting the patients were exposed to the same environmental source. However, no epidemiological information was available to investigate this possibility.

Secondly, using hypothesis-free population genetic methods, we identified that the CAPA isolates were a mixture of *A. fumigatus* genotypes. The genotypes from the 21 CAPA isolates showed genetic overlapping with the *A. fumigatus* genotypes isolated from the patients with IA, and those obtained from other patients and the environment. In the five previous genomic epidemiological studies of CAPA, the genotypes from the CAPA isolates were only compared with non-CAPA clinical isolates or COVID-19 patients who were colonised with *A. fumigatus* and had no active IA [21,22,23,24,25]. The results of Steenwyk et al. suggest that overall, the CAPA genotypes are drawn from the wider clinical population of *A. fumigatus* isolates, despite observing some genetic clustering [24]. This has been replicated in a larger transnational genomic study [21]. A larger multicentre study using microsatellite methods to genotype isolates identified that the genotypes obtained were genetically distinct from one another [23]. This finding was replicated in one Spanish and one Portuguese study using tandem repeats within the exons of surface proteins and *erg* coding genes (TRESPERG), and microsatellite methods, respectively, to compare isolates obtained from CAPA patients and the environment [22,25]. The authors identified that the CAPA isolates were from a diverse genetic pool [22,23,25]. However, compared to WGS, TRESPERG and microsatellite methods are limited in their power to provide an in-depth analysis of the genetic structure and diversity within a population [57,58], due to only utilising a small proportion of the whole genome [58]. Therefore, this study is the first to conduct an in-depth analysis to delineate the genetic diversity of *A. fumigatus* in CAPA cases. Furthermore, to show that the genotypes of *A. fumigatus* obtained from patients with CAPA are a genetically diverse mixture of clinical and environmental *A. fumigatus* genotypes, WGS is best placed to investigate [27,57].

COVID-19 patients who develop CAPA may have acquired *A. fumigatus* isolates from the environment. This is supported by the evidence that the genotypes of the CAPA isolates in this study are a mixture of genomes representative of both clinical and environmental *A. fumigatus* isolates. Previously, studies have identified that clinical non-CAPA *A. fumigatus* isolates are genetically similar to those sourced from the environment [22,25,32,56]. Additionally, AR*Af* has been isolated from several azole-naïve IA and CAPA cases [36,59,60] suggesting their resistance was pre-acquired from environmental (community and hospital) inocula. It has been hypothesised that patients who develop IA are first colonised with *A. fumigatus* isolates from airborne conidia in their environment. Thus, CAPA patients may acquire and become colonised with *A. fumigatus* on exposure to airborne *A. fumigatus* conidia in either the community or from the hospital environment. These conidia, under favourable conditions (immunosuppression from SARs-CoV-2 and drugs used to treat COVID-19, such as dexamethasone and tazoluzimab), develop hyphae and subsequently invade the surrounding tissue, leading to the onset of CAPA. If the patient is being exposed or treated with triazoles, it is at this stage that resistant inocula will remain in the patient’s airway.

The present study used bioinformatic tools and drug susceptibility tests to identify three AR*Af* out of 21 CAPA isolates containing the predominant *cyp51A* polymorphism TR_34_/L98H and that were phenotypically resistant to at least one azole. Furthermore, all three were pan-azole-resistant [36]. No further CAPA isolates had known azole-resistant polymorphisms and were susceptible to azoles phenotypically. A limitation of the current study is the screening tests to infer if an isolate is potentially susceptible or resistant. Thus, a robust and standardised approach (e.g., EUCAST broth microdilution method) is required to give the accurate and valid phenotyping of the isolates [26,39,41]. In a large Spanish tertiary hospital of 28 CAPA patients, the AR*Af* prevalence was zero [22]. This is despite AR*Af* being previously isolated from patients throughout Spain [61]. Similarly, a recently published small transnational study (*n* = 11) and a small German study (*n* = 4) did not identify any AR*Af* in the sample of CAPA isolates [21,24]. In a German multicentre study, the prevalence of AR*Af* isolates was found to be 22.2% (*n* = 6/27) [23]. Only one of the six CAPA isolates was identified to have known *cyp51A* polymorphisms conferring resistance [23]. In the Netherlands, a screening programme of CAPA patients identified one patient in a cohort of twenty-two to have an AR*Af* strain (4.5%) [62].

In the wider patient population, the prevalence of AR*Af* has been increasing in the last 30 years from 0.43% to 2.2% in London [63]. Different countries have recently reported varying levels: Germany at 3.5%, Denmark at 6.1%, Spain at 7.4%, Netherlands at 11.7%, and Japan at 12.7% [17,20,61,64,65]. In the UK, the prevalence of AR*Af* in the air was recently found to range from 3 to 9%, depending on the season [16,19]. Thus, the consideration of AR*Af* surveillance in CAPA and wider patient populations who may be exposed to *Aspergillus* conidia in the hospital air could be recommended owing to this near-ubiquitous exposure. However, this is not a cause for changing patient treatments.

There are some limitations to this study. During the COVID-19 pandemic, there were challenges in diagnosing CAPA. Therefore, relatively few patients were diagnosed with CAPA, so the sample size of this study was small. The small sample size of the CAPA isolates prevents the drawing of strong conclusions. Thus, future studies should include larger sample sizes. Secondly, it would have been useful to have clinical indicators and outcomes for the CAPA isolates to analyse any associations between the genotype and clinical phenotypes. This would help with understanding CAPA and improve patient outcomes. Finally, AR*Af* in CAPA was only identified in isolates from one country, owing to a small sample size. Thus, conclusions about the AR*Af* prevalence should be taken with caution. Ensuing studies with larger sample sizes from multiple countries across the globe would be better suited to determine the overall prevalence of AR*Af* in CAPA.

## 5. Conclusions

The present study is the largest transnational epidemiological analysis of the genomic relationship between CAPA isolates and the wider *A. fumigatus* population. We demonstrate that the genomes of the CAPA isolates comprised a diverse mixture of genotypes from the wider *A. fumigatus* population. The CAPA genomes are similar to those found in other clinical and environmental *A. fumigatus*, and are not composed of a unique sub-population. Finally, AR*Af* is identified in this at-risk group of patients.

Future directions should focus on developing user-friendly and easily accessible surveillance methodologies for assaying *A. fumigatus* for both clinical and environmental settings. Surveillance programmes should include methods of monitoring antifungal resistance, owing to the observed prevalence of resistance to frontline clinical azoles (e.g., itraconazole, voriconazole, isavuconazole, and posaconazole). Furthermore, exposure assessments could be used to aid clinical decisions on the most effective course of antifungal treatment. Finally, a review of local antifungal treatment guidelines in response to the increasing prevalence of AR*Af* needs to be further considered.

## Figures and Tables

**Figure 1 jof-09-01104-f001:**
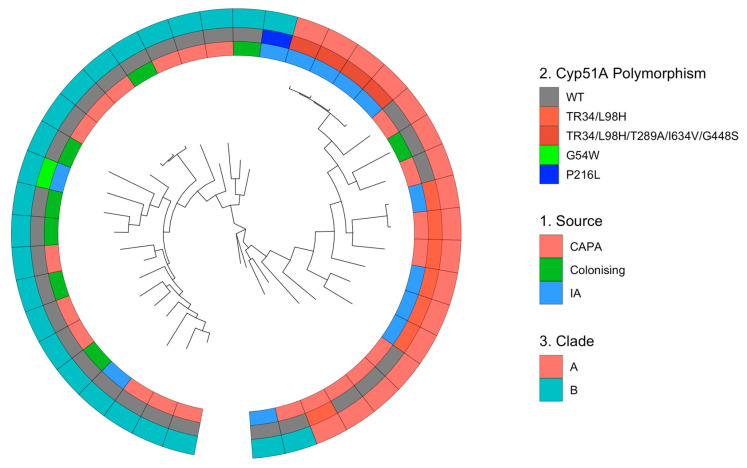
Phylogenetic tree of isolates from CAPA, IA, and colonising patients. Unrooted maximum likelihood (ML) phylogenetic tree over 1000 replicates performed on WGS SNP data, showing the following: inner track 1, the source of the isolate (CAPA, colonising or IA); middle track 2, if the isolate contains *cyp51A* polymorphism; and outer track 3, the clade in which the isolate is located.

**Figure 2 jof-09-01104-f002:**
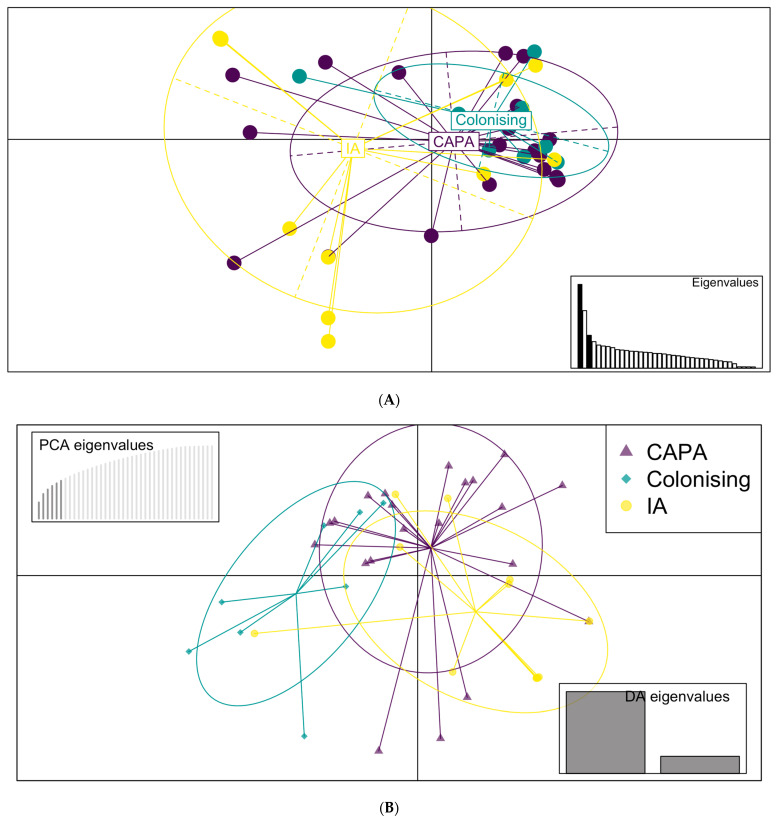
Multivariate analysis of isolates from CAPA, IA, and colonising patients. (**A**) Scatterplot of the principal component analysis (PCA) *A. fumigatus* genotypes using the first four principal components (PCs) illustrating the genetic identity of CAPA and IA control isolates. (**B**) Density plot of the discriminant PCA (DAPC), broadly identifying two distinct clusters, CAPA and control. (**C**) Composition plot highlights that all the isolates’ genotype is composed of genetic material identified as CAPA, IA, or colonising. Four isolates contain 70% control compared to CAPA, whereas there are 21 isolates the genome membership of which is mostly CAPA (>80%).

**Figure 3 jof-09-01104-f003:**
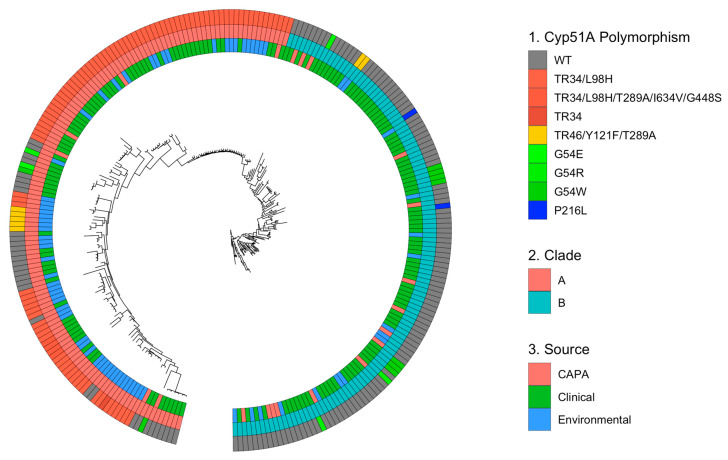
Phylogenetic tree of CAPA and *A. fumigatus* clinical and environmental isolates from Ireland and the UK. Unrooted ML phylogenetic tree with over 1000 replicates performed on WGS SNP data, showing the following: outer track 1, if the isolate contains *cyp51A* polymorphism; middle track 2, the clade in which the isolate is located; and inner track 3, the source of the isolate (CAPA, clinical, or environmental).

**Figure 4 jof-09-01104-f004:**
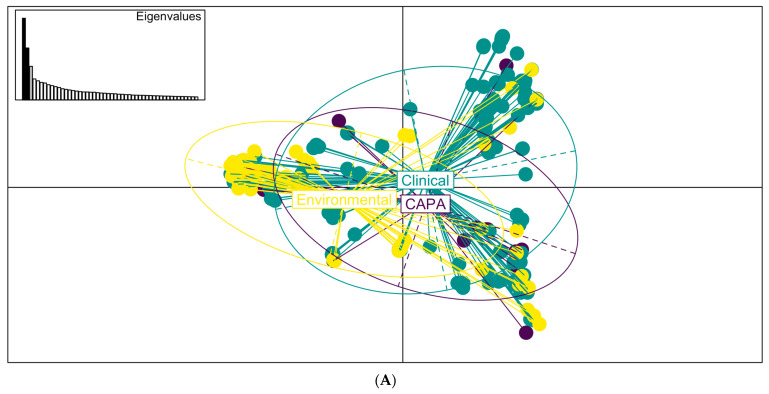
Multivariate analysis of CAPA and *A. fumigatus* clinical and environmental isolates from Ireland and the UK. Multivariate analysis, (**A**) scatterplot of the PCA *A. fumigatus* genotypes using the first five PCs illustrating the genetic identity of CAPA isolates, and clinical and environmental isolates from Ireland and the UK. (**B**) Scatterplot of the DPCA identifying 3 types form separate clusters, with CAPA and clinical with some overlap. (**C**) A composition plot comparing the genetic composition of each isolate. The plot highlights that each isolate is a mixture of genotypes identified as CAPA, environment, and clinical non-CAPA.

**Table 1 jof-09-01104-t001:** Characteristics of CAPA isolates of *A. fumigatus* used in this study and details of alignments.

Isolate ID	Country	Sample Type	No. of Aligned Reads (Millions)	Mean Depth of Coverage (x)	Percentage of Reference Genome Covered (%)
CAPA-A	Germany	CAPA—Possible	35.8	146.7	98.4
CAPA-B	Germany	CAPA—Probable	40.1	166.7	98.5
CAPA-C	Germany	CAPA—Probable	37.5	157.4	98.1
CAPA-D	Germany	CAPA—Possible	34.2	147.5	97.3
C403	Netherlands	CAPA—Possible	37.2	165.0	98.6
C408	Netherlands	CAPA—Possible	28.5	127.7	98.3
C422	UK	CAPA—Probable	11.8	45.5	96.2
C423	UK	CAPA—Probable	15.6	56.2	96.5
C424	UK	CAPA—Probable	11.2	45.1	98.6
C425	UK	CAPA—Probable	11.0	43.0	97.2
C434	Ireland	CAPA—Probable	10.6	41.9	96.6
C435	Ireland	CAPA—Probable	10.3	41.4	97.7
C436	Ireland	CAPA—Probable	9.8	39.3	98.2
C437	Ireland	CAPA—Probable	9.6	38.9	97.4
C438	Ireland	CAPA—Probable	10.5	42.6	97.7
C439	Ireland	CAPA—Probable	11.4	45.1	98.3
C440	Ireland	CAPA—Probable	7.8	27.6	97.9
C441	Ireland	CAPA—Probable	9.2	36.3	97.9
C443	Ireland	CAPA—Probable	9.7	39.2	97.5
C444	Ireland	CAPA—Probable	11.0	43.2	97.9
C611	UK	CAPA—Probable	11.0	43.7	97.6
C612	UK	CAPA—Possible	11.0	44.0	97.5

**Table 2 jof-09-01104-t002:** Azole drug susceptibility of CAPA isolates grown in minimal media and the associated candidate polymorphisms.

Isolate ID	VIPcheck^TM^ Score			Tebucheck Score	Resistance Marker
	ITR	VOR	POS		
C154	0	0	0	1	WT
C403 *	ND	ND	ND	ND	WT
C408 *	ND	ND	ND	ND	WT
C422	ND	ND	ND	1	WT
C423	ND	ND	ND	1	WT
C424	ND	ND	ND	1	WT
C425	ND	ND	ND	1	WT
C434	0	0	0	1	WT
C435	ND	ND	ND	1	WT
C436	0	0	0	1	WT
C437	0	0	0	1	WT
C438	1	1	1	3	TR_34_/L98H
C439	ND	ND	ND	1	WT
C440	0	0	0	1	WT
C441	1	1	0	4	TR_34_/L98H
C444	ND	ND	ND	3	TR_34_/L98H
C611	0	0	0	1	WT
C612	ND	ND	ND	1	WT
CAPA-A *	ND	ND	ND	ND	WT
CAPA-B *	ND	ND	ND	ND	WT
CAPA-C *	ND	ND	ND	ND	WT
CAPA-D *	ND	ND	ND	ND	WT

C154 was used as a control, as it had previously been shown to be EUCAST susceptible [32] and in this study, this confirmed by Tebucheck and VIPcheck^TM^. VIPcheck^TM^ score is 2 for uninhibited growth, 1 for minimal growth in well, and 0 for no growth. The wells contain either ITR 4mg/L, POS 0.5 mg/L, or VOR 2mg/L [40]. Tebucheck score of 1 = fully susceptible to tebuconazole (0mg/L); 2 = resistant, growth in 6 mg/L of tebuconazole; 3 = resistant, growth in 8 mg/L of tebuconazole; 4 = resistant, growth in 16 mg/L of tebuconazole [39]. A score of 0.5 was given if there was partial growth (10 to 50%) in a well [39]. ***** 6 Isolates did not have VIPcheck^TM^ or Tebucheck results. C403 and C408 had azole drug susceptibility using EUCAST broth microdilution method and CAPA A–D using the Clinical and Laboratory Standards Institute (CLSI) broth microdilution method (Table A5).

## Data Availability

All raw reads have been submitted to the European Nucleotide Archive (ENA) under Project Accession no. PRJEB60964. Two hundred and twenty-one raw reads of clinical non-CAPA and environmental *A. fumigatus* isolates have also been deposited under project accession no. PRJEB27135. The raw short reads of four German CAPA isolates have previously been deposited to the NCBI’s GenBank database under BioProject accession no. PRJNA673120.

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
