# Peer review of "Genomic Epidemiology Identifies Azole Resistance Due to TR34/L98H in European Aspergillus fumigatus Causing COVID-19-Associated Pulmonary Aspergillosis"

_jof, 2023, doi:10.3390/jof9111104_

Round 1

Reviewer 1 Report

Comments and Suggestions for Authors

The authors present an interesting manuscript on a topic not yet fully investigated. An enormous amount of laboratory and bioinformatic analysis work is recognized. Despite providing a first overview of the genetic relationship between isolates from CAPA patients and other A. fumigatus isolates, the limited size of the sample analyzed prevents drawing solid conclusions. The limitations of the study and future prospects should be described in more detail (some are mentioned in the conclusions). In addition to these considerations, I would like to make the following minor comments:

Section 2.3 should be a sub-section of section 2.2.

Lines 196-206: The scoring systems described in this manuscript differs from those described in the referenced papers.

Line 429: Where does these figures come from? 21 CAPA isolates were included in the study. In this sentence you mention 22 CAPA isolates, making the 78,6% of what?

The finding that there were not significant genetic differences between A. fumigatus isolated from CAPA and non-CAPA patients should be mentioned in the conclusions.

Author Response

Dear reviewer, 

Thank you very much for your helpful comments and suggestions. We appreciate them and have incorporated all the suggestions you have made into the final submission. 

To answer your question about line 429, thank you for spotting this mistake. It should be 15 isolates in clade B, therefore 71.5% of the isolates and not 78.6%. The sample was originally bigger. However, after the initial analysis and write up it was discovered that some of the isolates were not CAPA but were environmental. This led to a second analysis. However, this statement on line 429 was overlooked and should have been changed!

Thanks again for your helpful comments and suggestions

Kind regards,

Ben

Reviewer 2 Report

Comments and Suggestions for Authors

Dear authors

This is an interesting article. 

It called my attention that in the 21 studied patients no one was a proven case of CAPA.

Why do you incubate VIP check plates at 45°C?

The product instructions say:

VIP check™ results can be reliably read after one day if growth of Aspergillus fumigatus is visible, and in all isolates after 2 days of incubation at 35-37°C.

Author Response

Dear reviewer, 

Thank you very much for your helpful comments and suggestions. We appreciate them and have incorporated the suggestions you have made into the final submission. 

In answer to your question, we left the conidia for 48 hours at 45 degrees, because we found that at 24 hours there was incomplete growth in the control wells. In other words, the endpoint had not been achieved, so we left them for another 24 hours so that control wells had full growth. The reason that growth was slower was due to using a weaker concentration of spore suspension.

We have added this explanation to the methods section.

I hope this answers your question.

Thank you for noticing this discrepancy, so that we could make the methods section more clear.  

Kind regards,

Ben